# Refractory Chylothorax Secondary to Sizeable Azygos Vein Hemangioma: Tailored Multimodal Treatment of a Challenging Case Report

**DOI:** 10.3390/medicina59010091

**Published:** 2022-12-31

**Authors:** Paolo Albino Ferrari, Federico Fusaro, Antonio Ferrari, Alessandro Tamburrini, Giulia Grimaldi, Massimiliano Santoru, Sara Zappadu, Elisabetta Tanda, Sonia Nemolato, Simone Comelli, Roberto Cherchi

**Affiliations:** 1Division of Thoracic Surgery, Azienda di Rilievo Nazionale ed Alta Specializzazione “G. Brotzu”, 09121 Cagliari, Italy; 2Neuroradiology and Vascular Radiology Unit, Azienda di Rilievo Nazionale ed Alta Specializzazione “G. Brotzu”, 09121 Cagliari, Italy; 3Department of Thoracic Surgery, University Hospital Southampton, Southampton SO16 6YD, UK; 4Vascular Surgery Service, Department of Surgery, University of Cagliari, 09042 Monserrato, Italy; 5Department of Pathology, Azienda di Rilievo Nazionale ed Alta Specializzazione “G. Brotzu”, 09121 Cagliari, Italy

**Keywords:** azygos hemangioma, chylothorax, mediastinal mass, selective embolization, thoracoscopic surgery, multidisciplinary team, angiography

## Abstract

Background: Mediastinal hemangiomas are rare, and their etiology remains unclear. Most patients affected have no pathognomonic clinical symptoms, and the diagnosis is often incidental. Due to the paucity of the available literature regarding the management of this disease, the choice and timing of treatment remains controversial. Case presentation: Herein, we report the case of a hemangioma of the azygos vein arch in a 66-year-old woman who presented with dyspnea, chest discomfort, dysphagia, and weight loss. A simultaneous right chylothorax refractory to conservative management was found. A CT-guided biopsy of the mass was performed, and it confirmed the vascular nature of the lesion. Therefore, the patient underwent an angiography followed by endo-vascular embolization. Three days later, thoracoscopic surgical resection of the mass and the repair of the chyle leakage were performed safely. The patient was discharged uneventfully on postoperative day seven, with complete resolution of all the presenting symptoms. Conclusions: Treatment of symptomatic mediastinal hemangiomas could be mandatory, but a thorough multidisciplinary approach to these rare malformations is essential. Despite the risk of intraoperative bleeding, selective endovascular embolization followed by thoracoscopic surgery allowed for a complete and safe resection with a good outcome.

## 1. Introduction

Mediastinal hemangioma of the azygos (MHA) is a rare tumor that accounts for only 0.5% of mediastinal masses [1]. Histology classification entails three subtypes: cavernous, capillary, and venous [2]. Hemangiomas arising from the azygos arch are extremely rare, and very few cases have been described in the literature [3].

Venous malformations, tumors, portal hypertension, or local thrombosis could be related to the occurrence of these lesions. However, a specific etiology has not yet been identified [3,4].

Patients presenting with MHA are mainly asymptomatic, although clinical signs and symptoms may result from compression of nearby structures [5]. The optimal treatment strategy for MHA remains uncertain, though a multidisciplinary approach is suggested to prevent intraoperative complications and improve postoperative outcomes [6].

This case was reported to describe and discuss the strategy used for diagnosing and treating a rare arterio-venous capillary hemangioma of the azygos arch, with uncommon clinical presentation.

## 2. Case Presentation

A 66-year-old lady was admitted to the emergency medicine department of our institution due to worsening dyspnea associated with palpitations, dry cough, and dysphagia.

Past medical history included smoking habits, hypertension, and breast fibroid.

A computed tomography scan (CT) of the chest was performed. It showed the presence of a sizeable right-sided posterior mediastinal mass close to the azygos vein without a clear dissection plane, and an approximate size of 5.5 cm × 6.7 cm × 7.5 cm on axial plane. A large ipsilateral pleural effusion was also present.

In the first instance, a chest drain was inserted, resulting in drainage of milky fluid, confirmed to be chyle with the laboratory tests (cholesterol level was 66 mg/dL, and triglycerides level was 374 mg/dL).

The radio-metabolic study with 18-fluorodeoxyglucose positron tomography (FDG-PET/CT) showed a delayed enhancement in the central mass and faint uptake in its periphery with a standardized uptake value max of 0.7. Based on this evidence, a CT-guided biopsy was performed with uneventful results, and the pathology report was compatible with venous hemangioma (Figure 1).

Due to the severity of the symptoms and the persistence of high chylous drain output despite the fat-free diet with total parental nutrition, the case was discussed in a multidisciplinary team (MDT) setting to establish the best-tailored treatment option. The agreed strategy included an angiographic study for possible preoperative embolization treatment, followed by surgical excision.

A new enhanced CT scan revealed two intercostal arterial branches supplying the MHA (Appendix A), and the dedicated angiographic study showed a contrast-dye capture without reports of direct arteriovenous shunts. The embolization procedure was then performed through a micro catheterism of the main arterial feeder originating from an intercostal branch in the biaxial system (Glidecath^®^ 5 Fr. Cobra catheter and Progreat™ 2.7 Fr. by Terumo, Terumo Corporation, Tokyo, Japan), administering microparticles containing 250–500 µm of Polyvinyl Alcohol (PVA). Two platinum coils were then placed to complete the embolization (Figure 2).

Seventy-two hours after the procedure, the patient underwent right video-assisted thoracoscopic resection of the mass using a 4-ports approach. The choice of port placement depended on the posterior location of the mass: one port at the seventh intercostal space on the anterior axillary line, and two at the eighth intercostal space on the middle and posterior axillary lines, respectively. A fourth port was placed anteriorly at the fifth intercostal space to grasp and move the lung when necessary (Appendix A).

The esophagus was displaced anteriorly, and dense adhesions were present between the mass and descending aorta and between the mass and vertebral bodies. Several venous and arterial feeding vessels were controlled using a harmonic scalpel and titanium clip. The main pedicle connecting the mass to the azygos vein was divided with a standard mechanical stapler, and the mas was detached, preserving the azygos vein completely. The MHA was then removed into a 15 cm endo-bag through the anterior utility port (Appendix A).

Then, the chylothorax was carefully assessed, and a leak in the thoracic duct was found in the resected bed of the mass. This was interpreted as a direct injury caused by the compression of the lesion, and it was controlled with titanium clip ligation. To prevent a recurrence, chemical pleurodesis through pleural painting with iodopovidone solution was also performed [7]. An oxidized cellulose absorbable hemostat and a fibrin sealant were also injected over the residual surgical field for completion (Figure 3).

Final histology confirmed the diagnosis of capillary hemangioma (Figure 4).

The patient recovered uneventfully and was discharged home on postoperative day seven.

At one year follow-up, no symptoms were present, and the CT scan of the chest showed no evidence of MHA recurrence or pleural effusion (Appendix A).

## 3. Discussion

Despite the availability of reports on azygos vein arch malformations, the related subcategory of venous tumors is extremely rare, particularly capillary hemangiomas [3]. Very few cases of arterio-venous MHA have been reported in the literature and, to our knowledge, only a few instances of treatments have been comprehensively described.

Our case report is also the first to describe refractory chylothorax secondary to a decubitus injury on the lymphatic line as a complication of mass effect. Furthermore, given the high risk of intraoperative bleeding, we are the first to report the procedure of precautionary vascular embolization of MHA before surgical treatment.

The first report from Yamanaka et al. [8] underlined the challenges in diagnosing azygos vein malformations as a differential diagnosis of right posterior mediastinal masses.

In view of the various clinical presentation, MHA is often confused with fibromas, paragangliomas, lymphomas, and bronchial cysts [9,10,11,12]. Additionally, any suspicions of posterior mediastinal tumors with malignant biological behavior should be evaluated by diagnostic investigations, such as enhanced CT, magnetic resonance imaging, FDG-PET/CT, and digital subtraction angiography [13,14,15,16].

Preoperative tissue diagnosis by CT-guided biopsy or endoscopic ultrasonographic fine needle aspiration could validate MDT assessment in carefully selected cases when the non-invasive investigations are inconclusive [17,18]. In our case, we sought a tissue diagnosis before deciding on the best treatment. The worsening clinical condition caused by chylothorax led to weighing the risks of bleeding from the biopsy as lower than the surgical risks of a potentially open-and-close exploratory procedure.

Lipiodol-based percutaneous fluoroscopic lymphangiography can demonstrate whole lymphatic channels and intragenic leaks with a technical success rate of 75% to 100% [19]. Although our center also recognizes the usefulness of this examination as an integral part of the workup of the chylothorax, we did not perform it due to a lack of experience in this specific diagnostic investigation and the unavailability of the necessary equipment.

Mediastinal hemangioma is relatively benign, and therapies are still being debated. Moreover, there is a lack of guidelines or therapeutic flow charts that clearly define the indication for conservative, surgical, or endovascular treatment [6]. Therefore, we considered MDT mandatory, and the preoperative endovascular embolization procedure is a direct consequence of a multimodal approach.

Some cases of MHA have been successfully treated with embolization of the lumen with coils, covered stents implantation, or Amplatzer vascular plugs [20,21,22]. However, these procedures carry the risks of thrombotic migration and embolization, and a coil-filled MHA may produce worsening compressing symptoms from mass effect [23]. The treatment of choice should be driven by the significance of clinical symptoms and by the compression effects of the tumor on adjacent structures. Despite the tendency to bleed, complete surgical excision should be considered in patients with extensive and symptomatic MHA [24].

Interventional radiology can undoubtedly be helpful when the resection of a deep MHA becomes dangerous due to the inability to safely control bleeding, especially if an arterio-venous malformation is present [25]. This was evident in our case report, as the ability to use angiography to embolize major contributing vessels before surgery resulted in minimal intraoperative bleeding.

The decision on surgical approach is also affected by improved safety of hemostasis of minimally invasive surgery. Indeed, video-thoracoscopic contributed to excellent postoperative outcomes and enhanced recovery [1,3,6,8,16,18,25,26,27,28]. Unnecessary intraoperative manipulation of the MHA should be avoided to prevent hemorrhagic complications [29]. We efficiently completed a radical resection, and the video-assisted magnification allowed a precise detection of lymphatic leakage and its repair.

In our case, the prolonged postoperative stay was caused by common reactions to chemical pleurodeses, such as fever and the permanence of pleural drainage due to a large amount of serum output. As the reports in the literature also show [30,31,32], the efficacy of pleurodesis with iodopovidone was confirmed since the discharge by the absence of pleural effusion in short- and medium-term chest radiographies.

Although the complete resection of MHA has a good prognosis, and the disease is rarely malignant, a follow-up is needed within at least one year not to ensure no recurrence [33].

Treatment of vascular neoformations of the azygos vein and concomitant complications is often challenging and demanding. Preoperative planning must be thorough and discussed in an MDT to assess risks and benefits. Even in cases of complex arterio-venous MHA, thoracoscopic surgery preceded by selective vascular embolization techniques appears to be a safe and effective treatment.

## Figures and Tables

**Figure 1 medicina-59-00091-f001:**
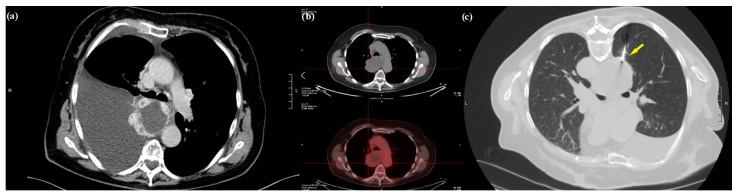
Preoperative diagnostic assessment: (**a**) chest CT scan showing a right pleural effusion and round-shaped mediastinal mass characterized by an irregular enhancing rim; (**b**) key image of FDG-PET/CT showing a delayed enhancement in the central mass and faint uptake in its periphery; (**c**) percutaneous CT-guided core needle biopsy of the mass (arrow) with a posterior approach. CT: computed tomography scan; FDG-PET/CT: 18-fluorodeoxyglucose positron tomography.

**Figure 2 medicina-59-00091-f002:**
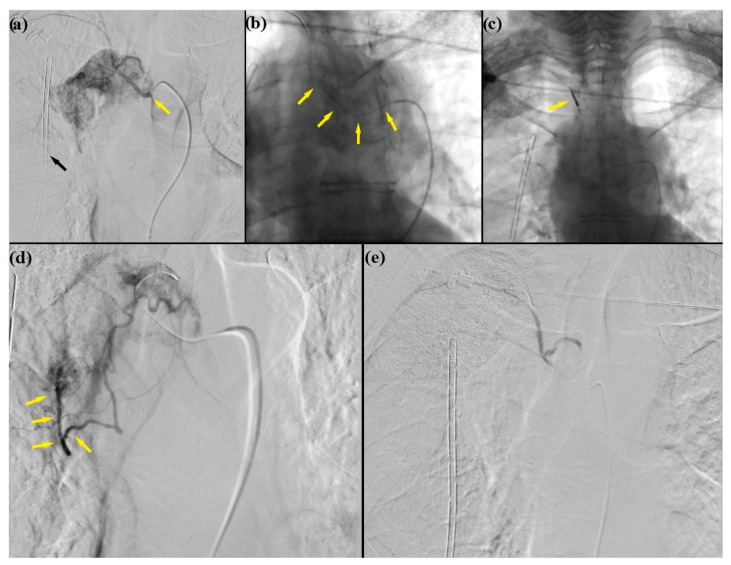
DSA and intra-arterial embolization: (**a**) catheterism (yellow arrow) of the right bronchial artery (Cobra Glidecath 4F, Terumo, Terumo Corporation, Tokyo, Japan) demonstrating a vascularized mediastinal mass and collateral image of the right pleural catheter (black arrow); (**b**) superselective catheterism (arrows) of an intercostal artery arising from the bronchial artery and; (**c**) subsequent embolization with micro coils (arrow) to avoid spreading of PVA particles; (**d**) superselective catheterism of main feeding artery and embolization with PVA particles (arrows) (Contour PVA 250–500 micron, Boston, Boston Scientific, Marlborough, MA, USA); (**e**) post-procedural DSA showing complete devascularization of the mass. DSA: Digital Subtraction Angiography; PVA: Polyvinyl Alcohol.

**Figure 3 medicina-59-00091-f003:**
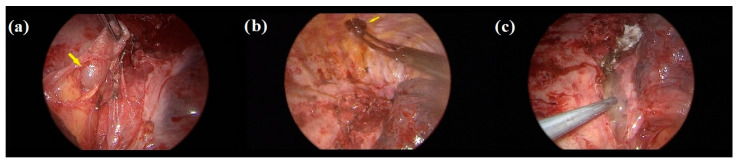
Thoracoscopic pitch images of the surgery, after complete hemangioma excision: (**a**) removal of the posterior thoracic gland (arrow) and duct clipping; (**b**) parietal pleurodesis with an iodopovidone-soaked sponge (arrow); (**c**) the residual surgical field is sealed with absorbable hemostatics.

**Figure 4 medicina-59-00091-f004:**
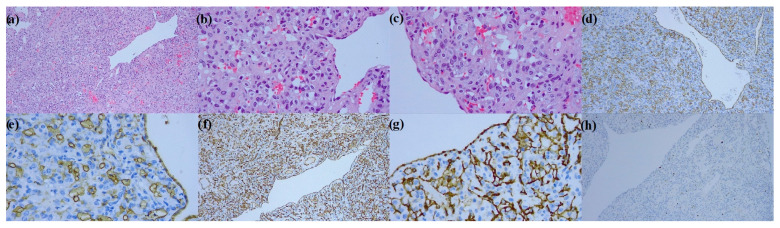
Pathology images: (**a**–**c**) hematoxylin and eosin stain: capillary hemangioma consists of small thin-walled vessels proliferation with lobular and solid growth patterns, dilated vessels, and more solid endothelial proliferation. Fibrosis, bland interstitial inflammation, and smooth muscle proliferation can be detected; (**d**,**e**) cytoplasmic immunoreactivity for CD31 in blood vessels and endothelial cells; (**f**,**g**) cytoplasmic immunoreactivity for CD34 in vessels layers and solid endothelial proliferation; (**h**) scattered nuclear positivity for ki67 in endothelial cells, very low proliferation rate is observed.

## Data Availability

Not applicable.

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
