# Peer review of "Refractory Chylothorax Secondary to Sizeable Azygos Vein Hemangioma: Tailored Multimodal Treatment of a Challenging Case Report"

_medicina, 2022, doi:10.3390/medicina59010091_

Round 1

Reviewer 1 Report

Abstract: Well written.

Introduction: Well written.

Case presentation: It would be interestin to mention the characteristics of the fluid that made it chylothorax (ie high TG, low cholesterol) - Line 60

Well described.

Discussion: The educational message of this case report and comparison with the literature are well presented.

Author Response

Thank you very much for taking the time to review our case report.

Regarding your comment, the milky pleural fluid laboratory tests were consistent with the macroscopic diagnosis of chylothorax. We have added the value of the laboratory tests in the text ("cholesterol level was 66 mg/dL, and triglycerides level was 374 mg/dL" line 59).

We performed a language revision in all the text and figure legends.

Thank you very much for your review.

Reviewer 2 Report

Line 60: the laboratory tests of the pleural effusion must be cited (especially cholesterol and triglycerides levels)

Line 61: How much was the SUVmax?

Any conservative treatment was attempted (MCT, TPN etc)?

Line 63: The CT biopsy was it uneventful?

Was a lymphography performed in order to evaluate the anatomy and patency of the thoracic duct and if not why?

Line 86: Where were the ports placed?

How much time was the duration of surgery?

How much were the blood loss?

Discharged in POD7 after a non complicated thoracoscopy does not sound as enhanced recovery. Please comment and cite the reasons for delayed discharge 

Author Response

Thank you very much for your time in reviewing our manuscript. All of your comments were very helpful in improving the article, and we hope that we have met your requests with the changes we have made to the text.

1) Line 59-60: the laboratory tests of the milky pleural fluid were coherent with the macroscopic diagnosis of chylothorax. We added laboratory tests value in the text.

2) Line 61-62: we added the SUV max value registered on the PET exam.

3) The patient underwent conservative free-fat dietary treatment and TPN without improvements. We added a comment in line 76.

4) The percutaneous biopsy was uneventful. We added this data in line 64.

5) Unfortunately, we did not perform lymphography as the appropriate equipment (contrast injectors) was unavailable. Generally, we require in-depth investigation in all patients who have then undergone surgery for refractory chylothorax.

6) The ports were placed according to Ivor-Lewis esophagectomy-like thoracoscopic approach. We added a brief description in the text, lines 96-100.

7) The total duration of the surgery "skin-to-skin" was 165 minutes. We felt it was not a striking result but relatively reasonable. That is why we did not specify the duration of surgery in the text.

8) No blood loss was reported intraoperatively, according to anesthesia and surgery records. With this data, we assume that the total blood collected during the operation was less than 100 mL.

9) In our case, the prolonged postoperative stay was caused by common reactions to chemical pleurodeses, such as fever and the permanence of pleural drainage due to a large amount of serum output. We added this comment in the text, lines 177-180.

We performed a language revision in all the text and figure legends.

Thank you very much for your review.

Round 2

Reviewer 2 Report

All issues were addressed sufficiently, however the comment about the non performance of lymphography should be added to the manuscript

Was chemical pleurodesis successful despite the high postoperative output? Please precise 

Author Response

Dear Reviewer,

Thank you very much for the in-deep revision of our manuscript. All your comments and suggestions have been seriously considered, and we updated the text to address your requests. 

1) we wrote comments about the non-performance of lymphography (lines 154-159) and added a new reference.

2) we extended the discussion with a comment on the efficacy of pleurodesis with iodopovidone (lines 186-188) and added three new references. 

Thank you very much for your interest in our manuscript.